# Child Nutrition Outcomes and Maternal Nutrition-Related Knowledge in Rural Localities of Mbombela, South Africa

**DOI:** 10.3390/children10081294

**Published:** 2023-07-27

**Authors:** Lucy Nomsa Masilela, Perpetua Modjadji

**Affiliations:** 1Department of Public Health, School of Health Care Sciences, Sefako Makgatho Health Sciences University, 1 Molotlegi Street, Ga-Rankuwa, Pretoria 0208, South Africa; 2Non-Communicable Disease Research Unit, South African Medical Research Council, Cape Town 7505, South Africa

**Keywords:** nutrition outcomes, nutrition knowledge, feeding practices, child–mother pairs, South Africa

## Abstract

Poor nutrition outcomes among children have become one of the major public health concerns in South Africa, attributed to poor feeding practices and maternal nutrition-related knowledge with conflicting data. In view of this, a cross-sectional study was conducted to determine the association of nutrition outcomes of children aged under two years with feeding practices and maternal nutrition-related knowledge in Mbombela, South Africa. Mothers’ nutrition-related knowledge was estimated using an adapted structured questionnaire on colostrum, continued breastfeeding, diarrhea prevention and treatment using oral rehydration solution, immunization, and family planning, and scored as excellent (80–100), good (60–79), average (40–59), and fair (0–39). This was along with questions on socio-demographic factors and obstetric history, as well as anthropometric measurements. Child nutrition outcomes were estimated by WHO classification using z-scores for stunting (length-for-age (LAZ)), underweight (weight-for-age (WAZ)), and thinness (body mass index-for-age (BAZ)). Using STATA 17, 400 pairs of children (8 ± 6 months) and their mothers (29 ± 6 years) participated in the study and were living in a poor socio-economic status environment. Half of children were stunted (50%) and over half (54%) were obese, while mothers were underweight (39%) and overweight (34%). In addition to one third of mothers reporting obstetric complications, two thirds, initiated breastfeeding within one hour of delivery, 30% exclusively breastfed, 48% introduced early complementary feeding, and 70% practiced mixed feeding. Twenty-eight percent (28%) of mothers had fair nutrition-related knowledge, while 66% had average knowledge, 6% good knowledge, and none of the mothers had excellent knowledge. A chi-square test showed that mothers’ nutrition-related knowledge was significantly associated with child stunting. The final hierarchical logistic regression showed significant associations of stunting with mothers’ nutrition-related knowledge (average: AOR = 1.92, 95%CI: 1.12–3.29), child’s age (6–11 months: AOR = 2.63, 95%CI: 1.53–4.53 and 12–23 months: AOR = 3.19, 95%CI: 1.41–7.25), and education (completing Grade 12: AOR = 0.36, 95%CI: 0.15–0.86). Contextual and intensified interventions on continued education for mothers to gain accurate information on nutrition-related knowledge and feeding practices could ultimately enhance child nutrition outcomes in poorer settings. Efforts should therefore be made to ensure that nutrition knowledge is appropriately provided based on the phases of child growth from 0 to 2 years, even beyond infancy into school age.

## 1. Introduction

Child nutrition outcomes (i.e., stunting, underweight, and thinness) are key to tracking the nutrition and health status of children in a population [1]. In particular, stunting (i.e., linear growth retardation or low length-for-age (LAZ)) is affecting 21.9% of children under five years, on a global level, and 41% in Sub-Saharan Africa (SSA) [2]. Stunting is common in the first two years of life and is an indicator of chronic undernutrition [3]. In developing countries, the predominance of stunting is attributed to dietary quality, which ultimately interrupts early linear growth in stages of life into adulthood, affecting body size and intellectual functioning [4,5]. In the long term, stunting among children predisposes them to short adult stature, reduced lean body mass, less schooling, diminished intellectual functioning, and overweight or obesity later in life as well as a higher risk of coronary heart disease, stroke, hypertension, and type 2 diabetes and unproductivity later in life [4,6,7,8,9]. 

In South Africa, child nutrition outcomes remain a public health concern [10,11,12,13]. Approximately 27% to 32.3% of children under five years in the country are stunted, followed by underweight (9%) and wasted (2.5%) [11]. Stunting, being a multifactorial phenomenon, is attributable to factors such as poor socio-economic status, poor maternal health and nutrition, inadequate infant and young child feeding (IYCF), and micronutrient deficiencies [14]. Appropriate IYCF entails the initiation of breastfeeding within one hour of birth, exclusive breastfeeding (EBF) for six months, a continuation of breastfeeding for up to two years and beyond, as well as timely introduction of complementary foods [15,16]. According to the South African Demographic and Health Survey (SADHS), the percentage of exclusive breastfeeding was 32% in 2016, while 19% of children (12–23 months) were breastfed. Recent studies have estimated stunting among children aged under five years as between 45.3% to 55% [17,18] and associated delayed introduction of solid foods with stunting [11] due to inappropriate food supplementation [17]. Other studies conducted in Burkina Faso [19] and Ethiopia [20] reported that stunting is worsened by breastfeeding continuing with proper complementary feeding specific for age, making infants beyond six months vulnerable to poor nutrition outcomes [21].

Several developing countries have adopted a multisectoral approach and implemented specific and sensitive nutrition interventions to reduce childhood stunting [22]. Further, most countries are prioritizing policies and programs that improve mothers’ ability to provide optimal care until the age of two years, a critical period for child growth and cognitive development and the origin of non-communicable diseases (NCDs) [22,23,24]. This is because the availability of parental resources in the first two years of a child’s life influences proper growth and cognitive development [3,4,5]. For instance, a mother’s knowledge in relation to food choices, feeding, health seeking, and her education has been studied [9,10,11,12,13,14,15]. According to the UNICEF Model of Care, the fundamental role of care in child nutrition indicates that caregivers require education, time, and support to provide adequate care [25]. Maternal knowledge and understanding of the aspects of basic nutrition and healthcare determine the care they give to their children [22,26,27]. It is worth noting that nutrition-related knowledge may be obtained from various sources [27], and if correct and put to practice, the nutrition-related knowledge may result in good child nutrition outcomes but, on the contrary, poor nutritional outcomes (i.e., stunting, underweight, and thinness) may result if incorrect information is put into practice [28]. Nonetheless, inconsistencies of the association of poor nutrition outcomes, nutrition-related knowledge, and feeding practices have been reported [29,30,31,32,33]. For instance, other studies reported significant associations [27,29,30], while others showed that adequate nutrition-related knowledge is not always translated into appropriate feeding practices [21,31,34,35]. 

The relationship of child nutrition outcomes and maternal nutrition-related knowledge in the South African context remains unclear. Few studies in the country have investigated several aspects of mothers regarding feeding knowledge [31], beliefs [36], practices [37], or dietary diversity [38] in relation to child nutrition outcomes. Nonetheless, stunting remains relatively high among South African children [17,18], and different forms of stunting and overweight/obesity in the same individuals [8,39], households, and settings coexist [38,40,41,42,43]. In view of this, the current study determined the association of nutrition outcomes of children aged under two years with feeding practices and maternal nutrition-related knowledge in Mbombela, South Africa. We hypothesize that in rural Mbombela, there is a significant relationship among feeding practices, nutrition-related knowledge, and child nutrition outcomes, especially with stunting. South Africa has promulgated several policies regarding feeding of children, including the Mother Baby Friendly Initiative (MBFI) and IYCF program [24,44]. Therefore, efforts to translate nutrition-related knowledge into appropriate action in an impoverished environment might be of help to mothers [45].

## 2. Materials and Methods

### 2.1. Study Design and the Conceptual Frameworks

This paper is an excerpt from a Master of Public Health dissertation written by the first author, which determined maternal feeding practices and nutritional knowledge, as well as the nutritional status of infants in community health facilities of the Mbombela sub-district in Mpumalanga Province. The entire study used a cross-sectional study design collecting data from mothers and their children under two years on anthropometry, obstetric history, socio-demographic status, feeding practices, and nutrition knowledge. The study was conducted from September 2021 to December 2021. From this umbrella study, the first published paper was “Evidence of Concurrent Stunting and Obesity among Children under two Years from Socio-Economically Disadvantaged Backgrounds in the Era of the Integrated Nutrition Programme in South Africa” [13]. The current paper is focused on child nutrition outcomes and maternal nutrition-related knowledge. 

The methods used in this study have been described in detail in the first paper [13]. In a nutshell, the study was anchored on the Bronfenbrenner’s social ecological model [46] and UNICEF conceptual framework for malnutrition [47]. In addition to inadequate dietary intake and disease, food access, household factors, poverty, maternal care practices, health services, the family context, and social values have been implicated in child nutrition outcomes [46,48,49,50]. Furthermore, nutrition-related knowledge of mothers influenced by living standards and the food environment, as an important component of their care and practices, is necessary although not adequate for optimal healthy child nutrition and positive behavioral change [51,52]. Lastly, data were further informed by the WHO core indicators for optimal IYCF [21,53,54].

### 2.2. Study Setting and Population

The study was conducted in Mbombela, a local municipality situated in Ehlanzeni District, Mpumalanga Province, described in detail in the previous paper [13]. Mbombela is the capital city of Mpumalanga Province in South Africa with a mixed rural and urban population [55]. In this area, cultural values and norms are highly important. The total population in this district is estimated at 695,913, and there are six community health centers (CHCs) and 31 primary healthcare (PHC) facilities [56]. The six CHCs were chosen based on the fact that they were situated in the rural sections of the settings with poor infrastructure. In such settings, several public health issues such as feeding practices and poor nutritional status are less studied. 

This study included children aged under two years and their mothers as a study population. Children were identified during visits to the selected facilities in the study setting for childcare services with their biological mothers and were further eligible to participate in the study after their mothers gave written consent. Children who were brought to these facilities by non-biological mothers were excluded from the study. We further excluded children aged above two years, reported to be sick and on treatment, as well as mothers who were sick with any medical condition that might have affected their feeding practices during the first 6 months prior to giving birth to the participating child, mentally unfit to be interviewed, and below 18 years. 

### 2.3. Sample Size and Sampling Technique

A minimum representative sample size was calculated using total population of 3000 children visiting CHCs [57] and 95% confidence level [58]. A sample size of 405 children paired with their mothers was obtained from the selected CHCs using systematic random sampling. Recruitment of children was conducted when mothers were in the queues for consultation with slips requesting their availability for interviews and anthropometric measurements. 

### 2.4. Data Collection and Tools

#### 2.4.1. Socio-Demographic Status 

Data were collected using a validated questionnaire administered by trained research assistants. The questionnaire was adapted from studies conducted with a similar research concept [8,17,18] to collect data on anthropometry, obstetric history, socio-demographic status, feeding practices, and nutrition knowledge from mothers and their children under 2 years. SiSwati-speaking independent translators conversant in English translated the questionnaire forward and backward between the two languages [59]. Content, construct, and face validity were ensured and approved by an expert committee. Following obtaining ethical approval from the Sefako Makgatho Health Sciences University Research and Ethics Committee (SMUREC) (SMUREC/H/290/2020: PG) and permission to access facilities in Mpumalanga Province (MP_202012_005), the study commenced. First, research assistants were trained to collect data, followed by a pilot study conducted among 30 participants to test the feasibility of the study and assess the efficiency of the research assistants to measure anthropometry and administer the questionnaire to participants. A reliability coefficient of 0.82 (i.e., Cronbach’s alpha) for the questionnaire was obtained. The results of the pilot study did not form part of the main study but informed improvement of the questionnaire in terms of the wording and structure. 

#### 2.4.2. Anthropometric Measurements

Anthropometry of both children and mothers was measured according to WHO procedures [60]. Seca 354 baby (Medicare, Germany) and smart D-quip electronic scales were used to measure weights (i.e., three measurements) of children and mothers to the nearest 10 g and 0.1 kg, respectively. Recumbent length (L) of children and height of mothers were measured using a Seca 210 measuring mat (Medicare, Germany) and stadiometer to the nearest 0.1 cm. Waist and hip circumferences were also measured from mothers using a non-stretchable tape measure to the nearest 0.1 cm. Identity number, date of birth, age, sex, weight, and height of children were captured on the WHO Anthro software version 3.2.2.1 and analyzed according to WHO z-score classification for length-for-age (LAZ), weight-for-age (WAZ), and body mass index-for-age (BAZ) [61]. Stunting (i.e., low LAZ/HAZ), underweight (i.e., low WAZ), and thinness (low BAZ) were defined at <−2SD, while overweight was defined by BAZ > 2SD, and obesity by BAZ > 3SD [60]. The calculated body mass index (BMI) defined underweight (<19 kg/m^2^), normal weight (19–24.99 kg/m^2^), overweight (≥25 kg/m^2^), and obesity (≥30 kg/m^2^), while abdominal obesity was defined as waist circumference (WC) ≥ 88 cm, waist–hip ratio (WHR) ≥ 0.85, and waist-to-height ratio (WHtR) ≥ 0.5.

#### 2.4.3. Mothers’ Feeding Practices and Nutrition-Related Knowledge

Feeding practices include early initiation of breastfeeding, exclusive breastfeeding under 6 months, continued breastfeeding at 1 year, introduction of solid, semi-solid, or soft foods, dietary diversity, minimum meal frequency, minimum acceptable diet, and consumption of iron-rich or iron-fortified foods (eight WHO core indicators of IYCF) [21,53,62]. 

Adapted from Fadare et al. [30], we used colostrum, continued breastfeeding, diarrhea prevention and treatment using oral rehydration solution, immunization, and family planning to assess mothers’ nutrition-related knowledge. Mothers’ ability to answer “yes” or “no” to a set of questions relating to child health and nutrition was assessed, and the nutrition-related knowledge was ascertained using frequencies and percentages of the correct, incorrect, or “don’t know” answers and categorized into excellent (80–100), good (60–79), average (40–59), and fair (0–39). 

### 2.5. Data Analysis

Descriptive and inferential statistics were computed using STATA version 17 (StataCorp. 2021. Stata Statistical Software: Release 14, College Station, TX, USA). Missing data were assessed using complete case analysis. Distribution of data was assessed using the Shapiro–Francia test. Results are presented as median (interquartile range (IQR)). A chi-square test (or Fisher’s exact test for cells with fewer than five cases) was used to compare child nutrition outcomes and selected feeding practices by maternal nutrition-related knowledge. Univariate (*p* < 0.20) and multivariate logistic regression (*p* < 0.05) analyses were used to determine the association of stunting with covariates, controlling for confounding. The main nutritional outcome of interest is child LAZ, which measures long-term nutritional outcomes, and association of stunting and covariates was determined using the crude odds ratio (COR) and adjusted odds ratios (AOR) and significance was considered at *p* < 0.05.

## 3. Results

### 3.1. Socio-Demographic Status

#### 3.1.1. Characteristics of Mothers

Information on socio-demographic factors, anthropometry, and obstetric history of mothers is presented in Table 1. The study consisted of mothers participating with their children (*n* = 400). The mean age of mothers was 29 ± 6 years with the youngest aged 15 years and the oldest aged 42 years. Two age groups were created in this study with 63% of women being younger (<30 years) and 39% being older women. BMI was defined as underweight (<19 kg/m^2^), normal weight (19–24.99 kg/m^2^), overweight (≥25 kg/m^2^), and obesity (≥30 kg/m^2^). Mean average BMI of mothers was 28.2 kg/m^2^ (±SD = 5) with a range of 19–39 kg/m^2^. Most mothers were underweight (39%) or overweight (34%). Abdominal obesity was defined as waist circumference (WC) ≥ 88 cm, waist–hip ratio (WHR) ≥ 0.85, and waist-to-height ratio (WHtR) ≥ 0.5. WC was 72 cm (±SD = 6.3) with a minimum of 64 cm and maximum of 100 cm and WHtR was 0.45 ± 0.37 (range: 0.39–0.60). Abdominal obesity was observed among mothers by WHtR (39%). Most mothers in this study were single (73%) and unemployed (83%) and 40% had attained secondary school education. Thirty-five percent (35%) of women lived in self-headed houses and 45% in parent-headed houses, while 76% lived in larger households (≥5 members). Household income was determined using monthly income from family members in South African rands (ZAR) and converted to US dollars (USD). Most mothers (30%) lived with their children in households with less than USD 5542/month (i.e., ZAR 5000/month) in income, followed by USD 5548–27,710 (i.e., ZAR 5000–ZAR 10,000) (27%). Parity of 1 to 2 was observed in 69% of women, and 30% reported having had obstetric complications. 

#### 3.1.2. Characteristics of Children

Table 2 shows the characteristics of children. Out of 400 children, there were 220 girls and 180 boys (mean age: 8 ± 6 months). Few children (*n* = 92) were aged 12–23 months, while 153 were aged 6–11 months and 155 aged 0–5 months. Low birth weight (LBW) was observed among 20% of children, while 49% were born at a normal weight, and 88% were born at a clinic (38%) or hospital (50%). 

### 3.2. Nutritional Outcomes of Children

Comparison of children’s nutrition outcomes by sex and age using a chi-square test (or Fisher’s exact test for cells with fewer than five cases) is presented in Table 3. WHO z-score classification and definitions for length-for-age (LAZ), weight-for-age (WAZ), and BMI-for-age (BAZ) were used to determine the nutrition outcomes: stunting (LAZ < −2 SD), underweight (WAZ < −2SD), overweight (BAZ > 2SD), and obesity (BAZ > 3SD). The prevalence of stunting (50%) was higher than that of underweight (1%) and thinness (4%) among children. In addition, the prevalence of obesity was high (54%), while overweight (13%) and overweight risk (13%) were also observed. No significant differences in stunting, underweight, overweight, and obesity were observed by sex. Significant differences by age were observed for all the nutritional outcomes among children. Children aged 6–12 months had a higher prevalence of stunting compared to their counterparts, while growth problems and obesity increased with age among children aged ≥12 months.

### 3.3. Associations of Stunting, Feeding Practices, and Nutrition-Related Knowledge

#### Mothers’ Feeding Practices and Nutrition-Related Knowledge

Child nutrition outcomes (dichotomized) and selected feeding practices are compared by mothers’ nutrition-related knowledge using a chi-square test (or Fisher’s exact test for cells with fewer than five cases) in Table 4. First, 63% of mothers, initiated breastfeeding within 1 h of delivery and 30% exclusively breastfed. Almost half of mothers (48%) introduced complementary feeding when their children were aged less than 3 months and 70% practiced mixed feeding. Twenty-eight percent (28%) of mothers had a fair nutrition knowledge, while 66% had average knowledge, 6% good knowledge, and none of the mothers had excellent knowledge. Mothers’ nutrition knowledge was significantly associated with stunting (*p* = 0.027) among children. 

Table 5 shows the relationship of stunting with independent variables. In univariate logistic regression, stunting was associated with average mother’s nutrition-related knowledge, child’s age, and education (i.e., completed Grade 12) (*p* ≤ 0.20). Multivariate analysis showed significant associations of stunting with mother’s nutrition-related knowledge (average: AOR = 1.92, 95%CI: 1.12–3.29), child’s age (6–11 months: AOR = 2.63, 95%CI: 1.53–4.53 and 12–23 months: AOR = 3.19, 95%CI: 1.41–7.25), and education (completing Grade 12: AOR = 0.36, 95%CI: 0.15–0.86). Therefore, average mother’s nutrition-related knowledge had a likelihood to predispose children to stunting, while the odds of stunting were high for children aged 6–11 and 12–23 months compared to those aged 0–5 months and completing grade 12 among mothers was protective against stunting in children compared to having no education. 

## 4. Discussion 

This study reports a high prevalence of stunting (50%) among children, accompanied by an alarming rate of obesity (54%). South Africa continues to experience persistent stunting affecting 22–55% of children [8,11,17,18,38,40], like in other African countries such as Ghana [29], Ethiopia [63], and Nigeria [30,63]. In addition to stunting predisposing children to poor cognition and educational performance as they advance in age [6], impaired fat oxidation associated with stunting makes children susceptible to obesity, including later in life [64,65]. Furthermore, a high prevalence of obesity among children in this study is consistent with other reports in South Africa [11,40,66]. Surprisingly, most women in this study were underweight (39%), like in Ethiopia (30%), Uganda (12%), and Tanzania (11%) [67], where high prevalence of underweight have been reported, which could be multifactorial. The prevalence of overweight (34%) reported in this study contributes to public health concerns in South Africa, although it is lower compared to maternal overweight/obesity in South Africa other studies, estimated at 54–76% [68,69,70]. High consumption of energy-dense food and reduced physical activity are always implicated in overweight/obesity in other countries [71,72,73], and in South Africa [41,68], nutrition transition is associated with frequent intake of processed and sugary foods as well as high-calorie and high-fat diets, socio-demographic characteristics, etc. [74,75,76,77].

Researchers have implicated chronic nutritional deprivation in stunting [78], usually associated with poor socio-economic conditions, poor maternal health and nutrition, frequent illness, and/or inappropriate infant and young child feeding and care in early life [79]. We attribute stunting in this study to household poverty, as indicated by the poor socio-economic status. Further, feeding practices, especially breastfeeding, are associated with socio-economic status, and in the current study poor feeding practices are characterized by a low level of exclusive breastfeeding and an early introduction of solid foods like in other studies in Africa [80,81,82] and locally [17,36,83]. On the other hand, children who consume formula milk become overweight/obese due to high protein and energy [66,67]. Furthermore, children who are susceptible to weight gain in the first week of life have a higher risk of weight gain between one and two years, and this doubles as they grow older [39,84,85]. During the first 6 months of life, exclusive breastfeeding protects children against overweight and obesity, among other benefits [13,48,86]. We attribute obesity in this case to poor feeding practices, such as early introduction of solid foods.

In this study, mothers and their children lived in socio-economically poor households and reported poor obstetric history, as also reported in similar South African studies [17,80]. Socio-economic status is one of the most important factors associated with health and medical outcomes of women of reproductive age [81,82]. In addition, women with lower socio-economic status are less likely to receive prenatal care, which is associated with poor obstetric outcomes [81,82,83]. Socio-economic differences across countries play a significant role [87] and are known to affect the prevalence of the double burden of malnutrition in LMICs, in addition to nutrition transition [42,43,88,89]. Therefore, the coexistence of maternal underweight and obesity, as well as child stunting and obesity, in poorer households suggests a household double burden of malnutrition [41,42,89]. Health implications of the double burden of malnutrition include increased mortality and morbidity risks, as well as poor cognition, health, and development with adult health risks [1,90]. 

The study has reported increased odds of stunting in children aged 6–11 months and 12–23 months. Increased odds of stunting with age have been reported among children in parts of Ethiopia [91], Indonesia [92], and Zambia [93]. Additionally, this study found average nutrition-related knowledge (66%) among mothers, similar to the 61.5% reported among caregivers in Ghana [94] and 59% in Kenya [95]. However, this knowledge does not seem to be reflected in practice since low levels of exclusive breastfeeding and early introduction of solids were observed, although two thirds were able to initiate breastfeeding within one hour in this study. It could be that in settings like our setting, a low level of parental education in terms of tertiary education might have no significant effect on child nutrition outcomes [60,96]. This kind of education status is unlikely to reinforce a mother’s nutrition knowledge for the purposes of reducing poor child nutrition outcomes.

On the contrary, high nutritional knowledge has been reported to be essential to promote good feeding practices and prevent poor nutrition outcomes among children in Kenya [97]. Our study reported a positive relationship between stunting and the mother’s education level, the same as in the study of Negash et al. [98] in Ethiopia and Fadare et al. [30] in Nigeria. A higher chance that a mother who completed Grade 12 would have a child who is less likely to be stunted compared to a mother with no education was observed in this study. It has been suggested that mothers with above a primary-level education can significantly reduce stunting in children [30]. Furthermore, the odds of mothers with average nutrition-related knowledge to have stunted children were 1.92 [29]. In Ghana, a positive link between a caregiver’s nutritional knowledge and stunting was reported among children aged 6–36 months [99]. Another study in Ghana, documented the effect of nutritional knowledge of mothers on children’s nutritional status and child feeding knowledge and practices were associated positively with appropriate feeding practices [96]. 

The significant difference in feeding practices of children in different stages of growth is well documented, and this requires mothers to have diverse nutrition knowledge to cater for those stages [30,100]. In this study, mothers were expected to recall their feeding practices from the outset after birth to the time the study was conducted. From our results, it is shown that the nutrition knowledge of mothers was not significantly different by children’s age group for the phases of exclusive breastfeeding in the first 6 months, introduction to solid food, as well as mixed feeding. This suggests a needed change in nutrition knowledge of mothers to feed their children appropriately as they advance in the various stages of growth, because their knowledge of food choices, feeding, and healthcare seeking is vital for producing normal child nutrition outcomes. For instance, as infants age beyond six months, they become more vulnerable to poor nutrition outcomes during the transition period from a milk diet to a diet that includes complementary food, predisposing them to stunting. This nutritional outcome is worsened by continued breastfeeding not accompanied by adequate complementary feeding at the appropriate age, due to nutritional demand [19,20,21].

Lastly, the context in which children grow up in can never be ignored as one of the influences on their nutrition outcomes, and the literature documents family context, ethnicity, and social values [12,101,102]. The impact of cultural beliefs and traditional practices is acknowledged in this study considering that the study setting is known for strong cultural values and norms, as well as poverty. First, the context of poverty in the current study, explained by poor living conditions, predisposes households to food insecurity through limited availability, access, and affordability and this will ultimately prevent children from meeting their nutrient requirements for growth [12]. Second, cultural beliefs and traditional practices contribute to feeding practices and vary across communities and countries [10,103,104,105]. Like in other countries, the role of mothers/caregivers regarding childcare is highly emphasized in most traditional cultures, where intergenerational influences on meal preparation inform systems and practices [105,106]. For instance, over two thirds of children in this study were non-exclusively breastfed and mixed fed. These are normalized feeding practices in other African societies with health impacts on children’s health due to exposure to the risk of diarrhea and malnutrition [5,102,107,108,109,110,111]. Furthermore, the pressure exerted by the family to give the baby other food further exacerbates the problem of early introduction of solids and other liquids [108,110], which might explain why almost half of the children in this study started to consume solid foods under 3 months of age. However, other researchers have acknowledged that some traditional practices, while often times misleading and not scientifically always sound, have been shown under certain circumstances to actually improve health/nutritional outcomes in children because they capitalize optimally on limited resources [105]. Therefore, in settings such as ours, challenges remain on how to assist mothers to make informed decisions regarding IYCF practices, considering culturally appropriate infant feeding messages. 

### Strengths and Limitations of the Study

Inferences made in this cross-sectional study are reliable, with limitations on establishing causality using this type of study design. We were able to obtain a random sampling, which mitigated selection bias. However, when it comes to collecting information from mothers, social desirability is the most concerning limitation since mothers might have withheld information or falsified their responses, especially on feeding practices, with the aim to be acceptable. We relied on self-reported information mostly, and having to remember previous events and practices might have also introduced recall biases. We cannot explain the high prevalence of underweight among women, as South Africa is known to have minimal levels of underweight among women of reproductive age and high levels of overweight/obesity compared to their counterparts. Conducting a health assessment of mothers including medical history might have provided a better explanation of underweight among mothers. Further, care should be taken when determining feeding practices; nonetheless, we used a WHO questionnaire, which was piloted prior to commencing the main study and entailed training research assistants on conducting interviews and anthropometry. Lastly, we did not measure the HIV status of mothers, which could have provided a better indication regarding their feeding practices, especially regarding breastfeeding. Also, confounding factors such as medication use and smoking and alcohol use as lifestyle factors should be considered in future studies. The results of this study are applicable to rural localities in Mbombela, Mpumalanga Province in South Africa and cannot be generalized to other areas because setting variations may influence feeding practices. Nevertheless, this study has reported on poor child nutrition outcomes, feeding practices, mothers’ nutritional knowledge, and their relationship. 

## 5. Conclusions

The high prevalence of stunting and obesity among children found in this study has great policy implications for South Africa, considering the lack of improved child nutrition outcomes, despite the introduction of the Integrated Nutrition Programme. Furthermore, poor feeding practices remains in South Africa even after having promulgated several policies regarding feeding of children. At the same time, poor maternal nutrition-related knowledge was prevalent, but most mothers had average nutrition knowledge. These findings contribute significant information for understanding the relationship between maternal nutrition-related knowledge and child nutrition outcomes in the context of a poverty-stricken environment. A significant association of stunting and mothers’ nutrition-related knowledge might be helpful in reducing stunting occurrences; however, this knowledge might not reflect good feeding practice. In addition, the association of mothers completing Grade 12 with lower odds of stunting among children could suggest that a low level of parental education in terms of tertiary education might not likely reinforce mothers’ nutrition knowledge for the purposes of reducing poor child nutrition outcomes. Therefore, contextual, and intensified interventions on continued education for mothers to gain accurate information on nutrition and feeding practices could ultimately enhance child nutrition outcomes in poorer settings. Efforts should therefore be made to ensure that nutrition knowledge is appropriately provided based on the phases of child growth from 0 to 2 years, even beyond infancy into school age. 

## Figures and Tables

**Table 1 children-10-01294-t001:** Socio-demography, anthropometry, and obstetric history of mothers.

Variables	Categories	Frequency (*n*)	Percentage (%)	Median (Range)
Age (years)	<30 ≥30	250 150	63 37	29 (15–42)
Weight (kg)				73.6 (44–119)
Height (cm)				161.5 (150–171)
BMI (Kg/m^2^)	Normal Underweight Overweight Obesity	105 155 135 5	26 39 34 1	28.2 (19–39)
WC (cm)	Normal Abdominal obesity	374 26	97 3	72 (64–100)
WHtR	Normal Abdominal obesity	243 157	61 39	0.45 (0.39–0.60)
Marital status	Single Married Divorced	292 56 52	73 14 13	
Education level	No education Primary school Secondary school Completed Grade 12 Tertiary	72 50 158 58 62	18 13 40 15 16	
Employment status	Employed Unemployed	83 317	21 79	
Household Head	Self Spouse Parents Grandparents Relatives	138 53 183 16 10	35 13 46 4 3	
Household income (USD/monthly)	<5542 5548–27,710 27,716–55,420 55,426–83,130 >83,130	61 61 109 93 76	15 15 27 23 19	
Parity	1–2 ≥3	277 123	69 31	
Number of children	1–2 ≥3	253 147	63 37	
Obstetric complications	No Yes	282 118	70 30	

BMI stands for body mass index, WC stands for waist circumference, WHtR stand for weight-to-height ratio, and USD stands for US dollars ($).

**Table 2 children-10-01294-t002:** Characteristics of children.

Variables	Categories	*n*	%
Sex	Boys Girls	180 220	45 55
Age (months)	0–5 6–11 12–23	155 153 92	39 38 23
Childbirth weight (g)	<2500 ≥2500	79 321	20 80
Childbirth order	First Middle Last	91 195 114	23 49 28
Place of birth	Home Clinic Hospital	47 153 200	12 38 50

g stands for grams.

**Table 3 children-10-01294-t003:** Comparison of children’s nutrition outcomes by sex and age.

Variables	All *n* (%)	Boys *n* (%)	Girls *n* (%)	*p*-Value
LAZ				
Normal Stunting Tallness	111 (28) 202 (50) 87 (22)	50 (28) 90 (50) 40 (22)	61 (28) 112 (51) 47 (21)	0.975
WAZ				
Normal Underweight Growth problem	267 (68) 3 (1) 121 (31)	116 (66) 2 (1) 57 (33)	151 (70) 1 (1) 64 (29)	0.595
BAZ				
Normal Thinness Overweight risk Overweight Obesity	61 (16) 16 (4) 51 (13) 50 (13) 213 (54)	21 (2) 10 (6) 22 (13 27 (15) 95 (54)	40 (19) 6 (3) 29 (13) 23 (11) 118 (55)	0.168
**Variables**	**0–5 Months** ***n* (%)**	**6–11 Months** ***n* (%)**	**≥12 Months ** ***n* (%)**	** *p* ** **-Value**
LAZ				
Normal Stunting Tallness	66 (43) 72 (46) 17 (11)	34 (22) 98 (64) 21 (14)	11 (12) 32 (35) 49 (53)	≤0.0001 *
WAZ				
Normal Underweight Growth problem	128 (88) 2 (1) 19 (3)	120 (79) 1 (1) 30 (29)	19 (21) 0 72 (79)	≤0.0001 *
BAZ				
Normal Thinness Overweight risk Overweight Obesity	42 (28) 4 (3) 27 (18) 21 (14) 55 (37)	10 (7) 12 (8) 15 (10) 18 (12) 96 (64)	9 (10) 0 9 (10) 11 (12) 62 (68)	≤0.0001 *

* Stands for significant difference, *n* stands for number of participants, IQR stands for interquartile range, LAZ stands for length-for-age z-scores, WAZ stands for weight-for-age z-scores, and BAZ stands for body mass index-for-age z-scores.

**Table 4 children-10-01294-t004:** Comparison of child nutrition outcomes and feeding practices by mothers’ knowledge.

Variables	All *n* (%)	Fair NK *n* (%)	Average NK *n* (%)	Good NK *n* (%)	*p*-Value
Stunting No Yes	111 (35) 202 (65)	41 (45) 50 (55)	59 (31) 130 (69)	10 (53) 9 (47)	0.027 *
Underweight No Yes	267 (98) 3 (2)	76 (100) 0	163 (98) 3 (2)	16 (100) 0	0.431
Thinness No Yes	62 (79) 16 (21)	19 (83) 4 (17)	36 (78) 10 (22)	6 (86) 1 (14)	0.849
Child overweight/obesity No Yes	88 176	19 (19) 82 (81)	35 (15) 203 (85)	6 (26) 17 (74)	0.291
LBW No Yes	321 (80) 79 (20)	85 (79) 23 (21)	205 (81) 48 (19)	20 (83) 4 (17)	0.823
Initiation of breastfeeding Within 1 h >1 h (delayed)	202 (63) 119 (37)	44 (56) 37 (44)	134 (65) 71 (35)	15 (75) 5 (25)	0.196
Exclusive breastfeeding No Yes	261 (70) 108 (30)	79 (77) 23 (23)	158 (69) 72 (31)	15 (65) 8 (35)	0.220
Mixed feeding No Yes	120 (30) 277 (70)	35 (32) 73 (68)	76 (30) 175 (70)	4 (17) 19 (83)	0.360
Introduction of complementary foods ≤3 months >3 months	144 (48) 158 (52)	41 (47) 46 (52)	86 (45) 104 (56)	11 (65) 6 (35)	0.306

NK stands for nutrition knowledge, LBW stands for low birth weight, * stands for significant difference.

**Table 5 children-10-01294-t005:** Association of stunting with covariates.

Stunting	COR (95%CI)	*p*-Value	AOR (95%CI)	*p*-Value
Mother’s nutrition-related knowledge Fair Average Good	1 1.81 (1.08–3.02) 0.74 (0.27–1.99)	0.024 * 0.548	1 1.92 (1.12–3.29) 0.63 (0.22–1.85)	0.018 * 0.403
Child’s age (months) 0–5 6–11 12–23	1 2.64 (1.58–4.42) 2.67 (1.24–5.71)	≤0.0001 * 0.012 *	1 2.63 (1.53–4.53) 3.19 (1.41–7.25)	≤0.0001 * 0.006 *
Mother’s education No education Primary Secondary Completed Grade 12 Tertiary	1 0.73 (0.31–1.74) 0.95 (0.48–1.86) 0.36 (0.16–0.80) 0.79 (0.34–1.85)	0.475 0.872 0.013 * 0.592	1 0.90 (0.36–2.27) 1.11 (0.54–2.28) 0.36 (0.15–0.86) 0.75 (0.30–1.85)	0.831 0.783 0.021 * 0.528

COR stands for crude odds ratio, AOR stands for adjusted odds ratio, and * stand for *p* < 0.05.

## Data Availability

The dataset for the study groups generated and analyzed during the current study is available from the corresponding author upon reasonable request due to ethical restrictions.

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
