# Peer review of "Child Nutrition Outcomes and Maternal Nutrition-Related Knowledge in Rural Localities of Mbombela, South Africa"

_children, 2023, doi:10.3390/children10081294_

Round 1

Reviewer 1 Report

Dear authors. Your article has improved but there are still some issues that need to be corrected before publication. They are as follows:

Key words should not be included in the text. With the exception of feeding practices, all are.

Line 150:  which is CHCs ? write the full text name before the abbreviation. 

Material and Methods: indicate if there is a positive report from a bioethics committee.

All chapter and subsection numbering is incorrect.

material and methods, results, discussion and conclusions have heading 2.

Within each of the chapters, also the subsections are incorrectly numbered. 

Table 1. Characteristics of the mothers. It should be self-explanatory.  Even though it is described in material and methods, the authors should report what are the cut-off points for abdominal obesity as measured by waist circumference or waist-height ratio? 

The WC has no units. Household measure also has no units (what is the currency?). 

Table 3 . It must be self-explained. Add that WHO references were used 

Author Response

REVIEWER 1

Dear authors. Your article has improved but there are still some issues that need to be corrected before publication. They are as follows:

Key words should not be included in the text. With the exception of feeding practices, all are.

Response: Kindly bear with us that we don’t understand this comment. You can elaborate further, and indicate the line you want us to revise.

Line 150:  which is CHCs ? write the full text name before the abbreviation. 

Response: Added – line 135-136

Material and Methods: indicate if there is a positive report from a bioethics committee.

Response: added in lines 164-167

All chapter and subsection numbering is incorrect.

Response: numbering of sections has been corrected through the manuscript.

material and methods, results, discussion and conclusions have heading 2.

Within each of the chapters, also the subsections are incorrectly numbered. 

Response: Corrected

Table 1. Characteristics of the mothers. It should be self-explanatory.  Even though it is described in material and methods, the authors should report what are the cut-off points for abdominal obesity as measured by waist circumference or waist-height ratio? 

Response: Information added

The WC has no units. Household measure also has no units (what is the currency?). 

Response: Added

Table 3 . It must be self-explained. Add that WHO references were used 

Response: Revised

Reviewer 2 Report

The manuscript submitted to Children by Masilela and Modjaji titled: "Child nutrition outcomes and maternal nutrition-related knowledge in rural localities of Mbombela, South Africa" is an interesting observational human study aiming to investigate the link between nutrition outcomes in children under 2yrs and maternal nutrition-related knowledge in specific setting(s) that of rural Mboombela in South Africa. This is an interesting study investigating an important topic with public health implications especially as related to child health, growth and development in an understudied and underserved population. 

The reviewer would like to provide a few points for consideration by the authors:

1. Consider providing more information on the rationale for selecting the particular locations for data collection/study execution.

2. There is a significant potential difference in the "0" - 2yr range in terms of feeding achievement for infants. For example a 6 month old infant is likely to be breast fed whereas a 1.5 year old one has been introduced to solids most likely. This creates potentially significant change in the need for nutrition knowledge while also makes the issue of access increasingly more critical as the child grows older. This is something that the authors may wish to discuss in the discussion. Also discuss how they normalized for the potentially difference in the need for nutritional knowledge given the varying age/need of children.

3. Some traditional practices while often times misleading and not scientifically always sound have been shown under certain circumstances to actually improve health/nutritional outcomes in children because they capitalize optimally on limited resources. Such a discussion may be interesting and would most likely strengthen the paper. Here is an article that may be found useful in this regard: Kristo, A.S.; Sikalidis, A.K.; Uzun, A. Traditional Societal Practices Can Avert Poor Dietary Habits and Reduce Obesity Risk in Preschool Children of Mothers with Low Socioeconomic Status and Unemployment. Behav. Sci. 2021, 11, 42. https://doi.org/10.3390/bs11040042.

4. Did the study control/normalize for the following confounding factors: medication(s) and or smoking?

5. Was marital status considered in the selection criteria?

6. BMI is an index and the kg/m2 denotes the way the index is calculated and does not allude to units (BMI does not have units), we do not measure surface area or force. Thus please omit the units from BMI.

Author Response

REVIEWER 2

The manuscript submitted to Children by Masilela and Modjaji titled: "Child nutrition outcomes and maternal nutrition-related knowledge in rural localities of Mbombela, South Africa" is an interesting observational human study aiming to investigate the link between nutrition outcomes in children under 2yrs and maternal nutrition-related knowledge in specific setting(s) that of rural Mboombela in South Africa. This is an interesting study investigating an important topic with public health implications especially as related to child health, growth and development in an understudied and underserved population. 

The reviewer would like to provide a few points for consideration by the authors:

  1. Consider providing more information on the rationale for selecting the particular locations for data collection/study execution.

Response: Added; lines 138-139

  1. There is a significant potential difference in the "0" - 2yr range in terms of feeding achievement for infants. For example a 6 month old infant is likely to be breast fed whereas a 1.5 year old one has been introduced to solids most likely. This creates potentially significant change in the need for nutrition knowledge while also makes the issue of access increasingly more critical as the child grows older. This is something that the authors may wish to discuss in the discussion. Also discuss how they normalized for the potentially difference in the need for nutritional knowledge given the varying age/need of children.

Response: We are not sure if we understood this comment. However, we tried to add information as per the advice in lines 374-388 of the discussion. Especially this point - “Also discuss how they normalized for the potentially difference in the need for nutritional knowledge given the varying age/need of children” – kindly clarify further on this point, beyond the explanation we have given in lines; 374-388, and additions in the conclusion; lines 456-458.

  1. Some traditional practices while often times misleading and not scientifically always sound have been shown under certain circumstances to actually improve health/nutritional outcomes in children because they capitalize optimally on limited resources. Such a discussion may be interesting and would most likely strengthen the paper. Here is an article that may be found useful in this regard: Kristo, A.S.; Sikalidis, A.K.; Uzun, A. Traditional Societal Practices Can Avert Poor Dietary Habits and Reduce Obesity Risk in Preschool Children of Mothers with Low Socioeconomic Status and Unemployment. Behav. Sci. 2021, 11, 42. https://doi.org/10.3390/bs11040042.

Response: This is appreciated. We added the information considering the suggested paper and other relevant papers.

  1. Did the study control/normalize for the following confounding factors: medication(s) and or smoking?

Response: the study excluded any mother and child who were sick – lines 140-148 . No lifestyle factors were considered in this study – added in the limitations of the study; lines 430-433

  1. Was marital status considered in the selection criteria?

Response: The selection criteria for the study is indicated in lines 140-148. No sociodemographic variable was singled out to select participants.

  1. BMI is an index and the kg/m2 denotes the way the index is calculated and does not allude to units (BMI does not have units), we do not measure surface area or force. Thus please omit the units from BMI.

Response: With this comment, we choose to differ. We have adhered to WHO standards of anthropometry and on how they always display BMI results. No BMI results have been displayed without mentioning the units (kg/m2) Therefore, we prefer to leave the units for BMI, same as in this paper we considered. “Socio-demographic and behavioural determinants of body mass index among an adult population in rural Northern Ghana: the AWI-Gen study”